# The Role of Wildlife and Pests in the Transmission of Pathogenic Agents to Domestic Pigs: A Systematic Review

**DOI:** 10.3390/ani13111830

**Published:** 2023-05-31

**Authors:** Iryna Makovska, Pankaj Dhaka, Ilias Chantziaras, Joana Pessoa, Jeroen Dewulf

**Affiliations:** 1Veterinary Epidemiology Unit, Faculty of Veterinary Medicine, Ghent University, Salisburylaan 133, 9820 Merelbeke, Belgium; pankaj.dhaka2@gmail.com (P.D.); ilias.chantziaras@ugent.be (I.C.); joana.cardosopessoa@ugent.be (J.P.); jeroen.dewulf@ugent.be (J.D.); 2Centre for One Health, College of Veterinary Science, Guru Angad Dev Veterinary and Animal Sciences University, Ludhiana 141004, India

**Keywords:** African swine fever, biosecurity, pig farming, risk factors, wild boars

## Abstract

**Simple Summary:**

The processes of globalization, human activities, and the growing popularity of outdoor ecological pork production contribute to increased interactions between domestic pigs and wildlife. Wild animals and pests can carry harmful pathogens that pose a threat to domestic pigs, resulting in economic losses at the local, national, and global levels. However, our understanding of the specific pathways through which these pathogens are transmitted between these animals is limited. This review has collected and examined information on the diversity and extent of pathogen spread from various wild animals and pests to domestic pigs. In the European context, our assessment showed that wild boars accounted for 80% of the documented pathogen transmission, followed by rodents (7%) and deer (6%). Insects, wild carnivores, wild birds, cats, and badgers had a lower representation in the studies. Farms with low biosecurity levels, particularly in extensive rearing systems, were identified as higher-risk farms. In general, 65.5% of the included studies supported possible risks and risk factors with quantitative data. Based on these findings, it is recommended to implement proper farm biosecurity measures, strong fences, and control programs for rodents, pets, and insects, particularly in high-risk areas. It is also crucial to monitor wildlife and raise awareness among farmers about the risks associated with disease transmission.

**Abstract:**

Wild animals and pests are important reservoirs and vectors of pathogenic agents that can affect domestic pigs. Rapid globalization, anthropogenic factors, and increasing trends toward outdoor pig production facilitate the contact between domestic pigs and wildlife. However, knowledge on the transmission pathways between domestic pigs and the aforementioned target groups is limited. The present systematic review aims to collect and analyze information on the roles of different wild animal species and pests in the spread of pathogens to domesticated pigs. Overall, 1250 peer-reviewed manuscripts published in English between 2010 and 2022 were screened through the PRISMA framework using PubMed, Scopus, and Web of Science databases. A total of 84 studies reporting possible transmission routes of different pathogenic agents were included. A majority of the studies (80%) focused on the role of wild boars in the transmission of pathogenic agents to pig farms. Studies involving the role of rodents (7%), and deer (6%) were the next most frequent, whereas the role of insects (5%), wild carnivores (5%), wild birds (4%), cats (2%), and badgers (1%) were less available. Only 3.5% of studies presented evidence-based transmission routes from wildlife to domestic pigs. Approximately 65.5% of the included studies described possible risks/risk factors for pathogens’ transmission based on quantitative data, whereas 31% of the articles only presented a hypothesis or qualitative analysis of possible transmission routes or risk factors and/or contact rates. Risk factors identified include outdoor farms or extensive systems and farms with a low level of biosecurity as well as wildlife behavior; environmental conditions; human activities and movements; fomites, feed (swill feeding), water, carcasses, and bedding materials. We recommend the strengthening of farm biosecurity frameworks with special attention to wildlife-associated parameters, especially in extensive rearing systems and high-risk zones as it was repeatedly found to be an important measure to prevent pathogen transmission to domestic pigs. In addition, there is a need to focus on effective risk-based wildlife surveillance mechanisms and to raise awareness among farmers about existing wildlife-associated risk factors for disease transmission.

## 1. Introduction

Wild animals, including pests, are important reservoirs of pathogenic agents [1,2]. Because of anthropogenic influences, wildlife–livestock interactions are increasingly more common and difficult to avoid, thus facilitating the introduction, reintroduction, and circulation of pathogens in various animal hosts [3,4]. This poses a challenge to animal and human health and production systems in today’s globalized world [5,6]. Deforestation, the establishment of farms at forest interfaces, encroachment into wildlife habitats, illegal marketing of wild animals, and the shift toward outdoor livestock production are anthropogenic factors that facilitate the transmission of wildlife-associated pathogens [7,8].

The pig industry is facing threats from various pathogenic agents on both a global and regional scale [9,10]. These agents can cause high morbidity and mortality rates, leading to the implementation of additional prevention and control measures [11,12,13]. However, these measures can also impose additional economic burdens on the industry [3,14]. A recent study by Niemi (2020) estimated the African swine fever (ASF)-associated losses of the European pig markets between 2010–2019 and found that ASF led to a reduction of the exports of pig meat by approximately 15% and to a cut in the production quantity by more than 4% in the first year after the outbreaks [15].

Many different wild and stray animal species can come into contact with pigs [5]. Mammals such as wild boars (*Sus scrofa*), deer (*Cervus elaphus*, *Capreolus capreolus*), badgers (*Meles meles*), wild carnivores (foxes (*Vulpes vulpes*), wolves (*Canis lupus*), and golden jackals (*Canis aureus*)) have been identified as potential sources of transmission of pathogenic agents to domestic pigs [1,3,16]. The presence of wild birds on farms has also been considered as a potential source of the introduction and spread of infectious agents [2]. Rodents (rats (*Rattus norvegicus*) and mice (*Mus musculus*)) are viewed as a threat to farm biosecurity and can serve as a link between wild fauna and pigs, thus facilitating pathogens’ transmission [2,17]. In addition, stray (or pet) cats (*Felis catus*) and dogs (*Canis lupus familiaris*) around pig farms can be carriers of several swine-associated pathogenic agents [11]. Finally, vectors such as ticks and flies might transmit some of the most important pathogenic agents to domestic piggeries [11,18].

Contact between all these wild and stray animal species and domestic pigs is possible in a multitude of direct or indirect ways, influenced by factors such as the density of wild animals, the interactions of wild animals and birds, the access to facilities, food, water, fomites, manure, carcasses, and other human and transportation-related factors [2,8,9,19]. Other transmission routes such as improper processing of fodder or contamination of the feed by carcasses of dead wildlife during the production process were found to play roles in past outbreaks in pig farms [5,20].

To effectively implement biosecurity measures, it is crucial to understand the various ways in which wildlife can come into contact with pig farms, and the potential role they play in the transmission of pathogenic agents [9,21]. Identifying the most significant risk factors and high-risk areas is essential for developing and implementing effective prevention and control strategies [5,11].

During the last few decades, several studies demonstrated different wildlife–livestock interfaces, but there are considerable knowledge gaps regarding the transmission routes of infectious agents from wild animal species to pigs [2,9]. Thus, the present systematic review aims to collect information on the role of wild, feral, stray animals, wild birds, rodents, ticks, and insects, in the transmission of pathogenic agents to pig farms in the European context.

## 2. Material and Methods

The systematic literature review follows the PRISMA (Preferred Reporting Items for Systematic Reviews and Meta-Analyses) guidelines. The PRISMA’s checklist for systematic reviews was completed and is available as Appendix A.

Literature searches were performed on 9–26 December 2022 using three online databases. PubMed^®^, Scopus, and Web of Science were the databases chosen as they provide adequate scientific sources for studies published in journals focused on natural, veterinary, and interdisciplinary sciences [22]. Only peer-reviewed publications written in the English language and published from 2010 onwards were included.

The systematic review was conducted to address the research question ‘‘What is the role of wild animal species and pests in the transmission of pathogenic agents to pig farms in Europe?’’. The term ‘pig farm’ was defined as pigs kept in all types of rearing systems: extensive (outdoor), intensive (indoor), semi-extensive, and backyard farms. The term ‘wild animal species’ includes all free-ranging non-domesticated animals such as wild pigs, wild birds, stray dogs, stray cats, rodents, other ungulates, wild carnivores, insects, and ticks that can live or move close to pig farms and can potentially transmit pathogenic agents [5]. The ‘role in the transmission of pathogens’ was considered as a result of any direct or indirect interaction between wild animal species and domestic pigs. This could include the shared use of resources, such as water, feed, or farmlands.

Search terms for pig farms, wild animal species, pathogenic agents, transmission pathways, and European countries were combined by the Boolean operators “OR” and “AND” to identify publications reporting the possible interface/interaction between any wild animals and domestic pigs in Europe (Figure 1). The details on the used search strings are provided as Appendix A.

All records were imported into the online tool Rayyan (https://www.rayyan.ai/ (accessed on 27 December 2022)). This tool was used to facilitate and organize the systematic literature review process [23]. The titles, abstracts, and full texts of the peer-reviewed papers were evaluated using the inclusion and exclusion criteria shown in Table 1.

Two co-authors conducted an independent full-text analysis of each publication using predetermined inclusion and exclusion criteria following the PRISMA guidelines. Articles were excluded if there was insufficient information in the methodology to decide if the criteria were met or not. Any disagreements regarding the eligibility of a manuscript were resolved through discussion between the same two co-authors.

The data collected consisted of details on the country and year where/when the study was carried out, type of pig farm, disease/pathogen(s) involved, host/vector/source, infection/disease prevalence among domestic pigs and wildlife, and study outcome. Data were extracted and compiled in a Microsoft^©^ Excel 2019 sheet. The flow of information throughout the systematic review is presented in Figure 2.

## 3. Results

A total of 1250 publications were screened and compiled into a search library. Altogether, 493 records were excluded through de-duplication. This resulted in 757 records selected for title and abstract screening. After this step, full-text screening was carried out on 326 records, of which 84 were concordant with the eligibility criteria applied. Results showed that Spain (*n* = 15), Italy (*n* = 11), Poland (*n* = 8), Switzerland (*n* = 8), Portugal (*n* = 6), and Germany (*n* = 5) were the top six countries with the highest number of publications. Figure 3 provides further details on the number of publications per country.

Approximately 65.5% (*n* = 55) of the included studies reported quantitative data on the possible risks/risk factors of pathogen transmission without providing evidence on confirmed transmission routes or outbreak tracing. Only three articles (3.5%) provided evidence on the transmission routes from wildlife to pig farms based on field data confirmed by genetic typing [24,25,26]. Furthermore, approximately 31% (*n* = 26) of the studies did not present quantitative data and only formulated hypotheses on possible transmission routes or risk factors and/or contact rate(s) considering wildlife as potential hosts/reservoirs/vectors of different pathogenic agents. Details on the selected studies are provided in Table 2.

### 3.1. Sources of Infection

Out of the 84 selected articles, 80% (*n* = 67) focused on the role of wild boars in the transmission of pathogenic agents. Studies involving the roles of rodents (rat, black rat, yellow-necked mice, mice) (7%), deer (red deer, roe deer, fallow deer) (6%), wild carnivores (foxes, wolves, and golden jackals) (5%), and insects (stable flies, house flies) (5%) in the transmission of pathogenic agents to pig farms were the next most frequent, whereas the roles of wild birds (waterfowl, diurnal birds of prey), ticks (soft ticks), cats (wild cat, stray cat), hares, and badgers were less explored (Figure 4). Eight papers described the roles of multiple wildlife species (Table 2). The possible pathogenic agents with their hosts/vectors are presented in Figure 5.

The results in Figure 5 highlighted the significant role of wild boars as a host for several pathogens such as African swine fever virus (ASFV) [37,56,57,64,65,66], foot-and-mouth disease virus (FMDV) [28], *Brucella suis* [26,47,72,90,101], hepatitis E virus [25,29,43], *Salmonella* spp. [84,109], *Leptospira* spp. [52], porcine reproductive and respiratory syndrome virus (PRRSV) [95], *T. gondii* [105], *M. bovis* [92], etc. The other host species that harbored a high number of pathogenic agents such as *Trichinella* (*spiralis*, *britovi*, *nativa*) [70], *Leptospira* spp. [63], *Yersinia* spp. [98], HEV [49], *T. gondii* [36,63] were rodents and wild carnivores [63,98].

### 3.2. Transmission Pathways

During the analysis of the literature, it was observed that various pathogens can be transmitted from wild animals (or birds) and pests to domestic pigs through multiple routes, as described below. Figure 6 and Appendix A provide a summary of the risk factors associated with both direct and indirect transmission routes from wild animals and pests to domestic pigs.

#### 3.2.1. Direct Transmission

Direct contact between domestic pigs and wild animals can lead to the transmission of diseases and can occur in various settings, such as outdoor pig rearing systems, backyard piggeries, and situations where pigs have access to forests, woodlands, or abandoned pastures/fields [56,60,77,86,100]. This risk is particularly high in areas with a high-density wild boar population [24,27].

Poor biosecurity measures, such as inadequate fencing, can increase the risk of direct contact between domestic pigs and wild animals, especially wild boars [37,41,70]. Studies suggest that the frequency of pathogen transmission from wild boars to domestic pigs strongly depends on the population size and density of wild boars, with diseases such as ASF mainly transmitted through direct contact with infected animals [38,66,76]. Studies conducted in Romania on backyard pigs highlight the risk of ASF outbreaks through direct contact with wild boars [76]. In addition, for *Salmonella* spp. the risk of transmission is seen to be higher in areas with a high density of pig farms, especially in areas with wild boar habitats and free-range farmed pigs [105]. There are also documented cases of transmission of tuberculosis between wild boars and free-ranging pigs in Spain through direct contact [93]. Additionally, research conducted in Lithuania indicated that direct contact is the most effective way of PRRSV transmission from wild boars to domestic pigs [61]. Studies also observed the transmission of HEV infections from wild boars to domestic pigs, which can potentially happen within extensive rearing systems as reported from Northern Germany and Poland [43,67]. Contamination of pastures with infected aborted wild boar fetuses is another potential risk of transmission of porcine brucellosis between wild and domestic pigs [47,48].

Furthermore, owing to the absence of fencing, contact with other wildlife species, such as wild carnivores (foxes, wolves, jackals) and stray (or wild) cats has been observed and is described as source of transmission of *Trichinella* spp. to pigs [70,80]. In open farming areas where wild ungulates cross the farm boundary regularly, deer play a leading role in the transmission of *M. bovis* and *M. tuberculosis* [92,94]. In addition, in the United Kingdom, the European badger (*Meles meles*) has long been recognized as the major wildlife reservoir for *M. bovis*, suggesting that pigs raised outdoors or on holdings with poor biosecurity might be more vulnerable to infection [107]. However, direct contact between domestic pigs and infected wild animals can occur if the fence is improperly structured [38].

Direct transmission of pathogens from wild birds to pigs can occur through mechanical means, particularly in backyard farms where interactions between birds and pigs are possible. This transmission has been described for Influenza A viruses and *Salmonella* spp. especially in the absence of grids or nets on windows and doors [97,106]. Additionally, migrating waterfowl have been identified as potential carriers of *Brachyspira hampsonii*, which can infect pig populations, particularly in outdoor herds where grids or nets are not in place [91]. In Slovakia, diurnal birds of prey have also been linked to the direct transmission of *Trichinella* spp. to pigs. Based on epidemiological investigation on the farm, these birds potentially could have fed on infected pig carcasses and subsequently introduced the parasite to the farms [83].

Rodents may transmit pathogens to pigs through direct contact, the fecal-oral route, and accidental consumption by pigs [27,36]. Studies from Italy and Croatia have reported direct or indirect transmission of HEV from rodents to pigs because of the contamination of the farm environment with rodent feces [31,49]. A study from the Netherlands showed direct transmission of *Leptospira* spp. from rodents to domestic pigs [63]. Regarding *Yersinia* spp., a study documented the likelihood of fecal-oral transmission of the pathogen between rodents and pigs [98]. Additionally, a study from Denmark showed the transmission of *T. gondii* from mice to pigs through direct contact on farms, especially in open feed systems where feed leftovers attract mice [36]. Rodents can also play an important role in the transmission and/or maintenance of *T. spiralis* in domestic pigs [70].

Direct pathogen transmission through vectors is a significant concern, as it can facilitate the spread of disease among pig populations. Flies and ticks have been identified as vectors of various pig pathogens [33,35]. For example, a study from Lithuania demonstrated that ASFV can be transmitted through the ingestion of blood-sucking flies that had fed on infected wild boars [58]. On Danish farms, mechanical transmission of ASFV was observed during feeding by *Stomoxys calcitrans*, with the virus detected in the flies up to 72 h after feeding on infected blood [33]. ASFV DNA has also been found in insects (*S. calcitrans*) collected from farms without outbreaks, indicating the possibility of contamination from wild boars or neighboring farms [58]. Moreover, *Lucilia sericata* may be able to disperse the virus, as they can fly around 2 km in a few days [58]. Insects can facilitate virus transmission to domestic pig farms with inadequate hygiene and biosecurity, as observed in Poland during ASF outbreaks [66]. Additionally, *Ornithodoros* spp. ticks can be persistently infected with ASFV and act as a reservoir, playing an important role in transmitting the virus from wild boars to domestic pigs [73,74,75].

#### 3.2.2. Feed/Swill and Water

In areas with mutual habitats, such as during the acorn season, searching for food can lead to overlapping between wild and domestic pigs [92,107]. In Romanian backyard pig farming, the risk of ASF outbreaks has been observed during scavenging or contact with contaminated feed [76]. Organic or free-ranging pig production systems have been associated with an increased risk of *Trichinella* spp. because of contaminated pastures since pigs can be infected by feeding on carcasses or the offal of hunted or dead wild boars [83]. HEV infections in pigs have been reported to occur through exposure to contaminated water or feed [51,109]. Both direct and indirect contacts have been reported in the case of *E. coli* transmission between wild and domestic pigs, such as through sharing a similar environment contaminated with feces, water holes, or feeding sites [40,41]. The fecal-oral route is the common transmission route for salmonellosis, which can occur through the ingestion of contaminated feed or water with *Salmonella* spp. [85]. The frequent spreading of pathogenic agents by cats, especially when they are used for rodent control on pig farms, has been reported [69]. Cats having access to pig pens, water, and food have also been associated with the shedding of oocysts of *T. gondii* [96].

#### 3.2.3. Airborne/Aerosol Transmission

Close contact with wild boars can transmit diseases through aerosols, including ADV, brucellosis, and *M. hyopneumoniae* [88,89,104]. For example, in Spain, ADV was found to be transmitted through both direct contact with infected wild boars and the aerosol route over long distances [89]. Similarly, swine brucellosis may also be transmitted through aerosols, as reported in a study from Switzerland [101]. The airborne transmission of *M. hyopneumoniae* between farms and over long distances in areas of high farm densities is also well-documented [104], highlighting the potential risk of infection for domestic pigs through the aerosol route [46,97,103].

#### 3.2.4. Manure/Feces

In addition to the direct transmission of HEV through the fecal-oral route, contaminated pig manure used as fertilizer on agricultural land could also potentially lead to the transmission of salmonellosis and HEV to the wild boar population, which could then act as a reservoir and spread the virus to other pig farms [43,49,67]. This highlights the importance of proper management and disposal of pig manure to prevent the spread of pathogens and the need for coordinated biosecurity measures between pig farms and surrounding agricultural areas to minimize the risk of disease transmission [49]. Wild birds in the United Kingdom have been reported to transmit *Salmonella* spp. to pigs through the contamination of the farm environment with fecal droppings [106].

#### 3.2.5. Venereal Transmission

Venereal transmission occurring during mating was considered to be an important pathway of ADV infection in a study in East Germany [42]. Direct contact with wild boars is a common mode of transmission of brucellosis among domestic pigs in extensive grazing systems, including approaches of sexually active wild boars outside the fence, intrusions, and mating, especially if they share the same habitat, as observed in Italy, Spain, and Switzerland [72,90,101,102]. Although not common, studies from areas with semi-extensive breeding systems (such as in Tuscany or Sardinia) suggest that there is an increased risk of transmission of leptospirosis from wild boars to domestic pigs through mating [52].

#### 3.2.6. Indirect Transmission

Indirect transmission of disease from wild species present in the farm environment can occur through various human activities such as veterinarian visits, interactions with farmers, professionals from the pig sector, hunters, workers, and visitors [27,37]. Additionally, the transmission may also occur through contact with contaminated fomites, such as clothing, boots, and equipment, and contaminated vehicles [64,66,102]. Studies have shown that ADV can also be transmitted through indirect contact with infected wild boars, such as oronasal transmission via contaminated fomites, as observed in Spain [88,89]. Further, the use of contaminated bedding materials, urine, and nasal and mouth discharges have been found to be associated with outbreaks caused by ASFV, ADV, HEV, *Salmonella* spp., and *Leptospira* spp. [88,106]. Other sources of contamination include virus-contaminated tissues of fallen wild boars, contaminated carcasses, contaminated vegetal products, grass or crops, contaminated water, soil, and areas fertilized with contaminated pig manure [27,64,100].

## 4. Discussion

Publications on the topic of wildlife–livestock interactions and pathogen transfer mainly originated from southwestern and central Europe where outdoor pig production systems are more prevalent [13]. High pig densities in regions such as Denmark, Belgium, France, Italy, Poland, and Spain have also led to increased interest in research on this topic [13]. It is remarkable to notice that only 3.5% of the selected articles provided evidence on the transmission routes from wildlife to domestic pig farms based on genetic typing. Most articles reported on possible risks/risk factors of pathogen transmission without providing full-bodied evidence on confirmed transmission routes or outbreak tracing. This clearly indicated that there is room and a need for more quantitative epidemiological studies that describe the circulation and transmission of pathogens at the wildlife–livestock interface [110,111,112]. Additionally, there is a pressing need to incorporate new technologies such as GIS (Geographic Information System) and remote sensing for comprehensive studies on the epidemiology of contagious diseases [113,114,115,116].

The majority of studies (80% (*n* = 67)) focused on potential transmission routes and risk factors associated with wild boars in the vicinity of domesticated pig farming. Therefore, the main transmission route is considered direct contact with wild animals. Risk factors such as poor or inadequate biosecurity (especially lack of proper fencing) and farms located near high-density wild boar areas provide more opportunities for direct contact with wild boars, hence increasing the probability of spillover of different pathogens [37,38,65,79]. In addition, practices such as raising pigs in outdoor production systems increase the likelihood of interactions with wild boars and other wildlife such as deer, wild carnivores, badgers, and wild birds as well as with pests, necessitating the prioritization of interventions in high-risk regions [86,87,101,117]. This also emphasizes the need for the implementation of external biosecurity measures that are effective in preventing the access of wildlife species to pig farms [42,60,118,119].

To prevent direct contact and disease transmission between wildlife species and domestic pigs, on-farm biosecurity practices are crucial [21,120,121]. Main protective measures such as fencing, confining pigs in wild animal-proof structures, and species separation at water points can be effective [9,86,94]. Wire-mesh fencing with an electrified wire is effective but expensive [86]. Moreover, simple fencing can also reduce interaction risks. According to EFSA 2021, installing single or double fences on outdoor pig farms in ASF hotspot regions could reduce risk by 50%. On the other hand, the observations from past studies indicated that even with electric fences (e.g., pasture run-out), the outdoor pig farms located in forest regions are at risk of interactions with wild boars [86,101].

Studies have identified direct contact with wild birds as a common risk factor for the spillover of various pathogens. Specifically, the absence of grids and the ability of birds to access stables or other indoor areas can increase the likelihood of disease transmission. To mitigate these risks, it is important to implement effective management practices such as covering feed and water sources, using bird-proof netting or grids on windows and doors, and regularly cleaning and disinfecting the environment [106,122]. For pests, the most important gaps were linked with poor implementation of rodent and vermin control programs [79,83,123]. Thus, rodent management activities need to be carried out more effectively at the farm level. The use of domestic cats (which can carry *T. gondii* and other pathogens) for pest control is not a good practice [96]. In addition, cats and dogs should be kept out of the animal facilities owing to the possibility of pathogen(s) transmission [11,124]. The use of rodent-proof buildings is crucial when planning new facilities for production animals [17]. In tick-endemic regions, tick-proof housing should be constructed, and effective pest control and disinfestation procedures should be implemented to reduce the risk of pathogen circulation [75]. Indirect transmission through unsafe contact with the outside farm environment, such as the lack of physical barriers, poor disinfection, and personal biosecurity procedures, also poses a risk [37,41,93,125]. The role of contaminated litter in the transmission of pathogens should be accounted for during the development of biosecurity protocols. For example, the storage of feed and straw for certain periods is recommended for preventing the spread of ASF [27]. Moreover, contaminated manure was found to be linked with the spread of salmonellosis and HEV among pigs [109]. Therefore, appropriate treatment of manure is required to reduce the risk of pathogen spillover and environmental pollution [126]. Safe carcass management on farms is also crucial to avoid pathogen contamination and persistence [41]. The use of contaminated carcasses as feed can lead to *Trichinella* spp. and *T. gondii* infections in pigs, while wild boars infected with these pathogens can migrate and infect other animals [69]. Therefore, appropriate carcass control and disposal should be included in farm biosecurity protocols [13,124]. Furthermore, it is important for hunters to properly dispose of hunting remains by incinerating them to prevent the risk of infection to omnivores and carnivores [66,127]. Other protective measures include avoiding the feeding of uncooked pork and maintaining a closed herd [13,21,27]. Human-related activities such as the illegal sale of ASF-infected and wild pigs can also contribute to pathogen(s) spread, making the regular monitoring of illegal wildlife trade crucial [27,66].

All the above listed transmission routes clearly illustrate the importance of implementing good biosecurity [11,128]. To achieve this, a good understanding and awareness of the risks and their pathways are crucial [27,56]. In order to gain this valuable insight, the implementation of monitoring and surveillance systems in wildlife populations is needed [113,129]. These systems enable the swift detection of endemic and exotic diseases, as well as the enhancement of disease awareness among stakeholders [65]. In addition, effective monitoring of the implementation of biosecurity measures is essential to ensure their ongoing effectiveness [21,56,124].

## 5. Study Limitations

Although we attempted to minimize the limitations inherent in systematic reviews, such as publication bias and language restrictions, there may be additional relevant studies that were not included in this review. However, we believe that the evidence analyzed in this review is sufficient to support the themes and results presented above.

## 6. Conclusions

Based on a systematic review of the literature, wild boars are the main species transmitting pathogenic agents to domestic pigs in the European context. Other important vectors and reservoirs of infectious diseases include wild ungulates, rodents, wild carnivores, insects, wild birds, ticks, stray (or pet) cats, badgers, and hares. Outdoor farms or extensive systems and farms with low biosecurity are associated with a higher risk of pathogen introduction. To protect pig herds from the introduction of pathogens, it is essential to adopt appropriate farm biosecurity, proper fencing, and rodent, pet, and insect control programs, especially in high-risk regions. Additionally, there is a need for monitoring and surveillance programs for wildlife and raising awareness among farmers about wildlife-associated risk factors for disease transmission.

## Figures and Tables

**Figure 1 animals-13-01830-f001:**
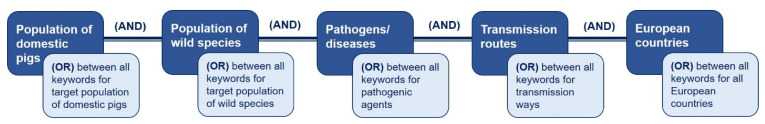
Outline of text strings used for the searches conducted in PubMed^®^, Scopus, and Web of Science databases.

**Figure 2 animals-13-01830-f002:**
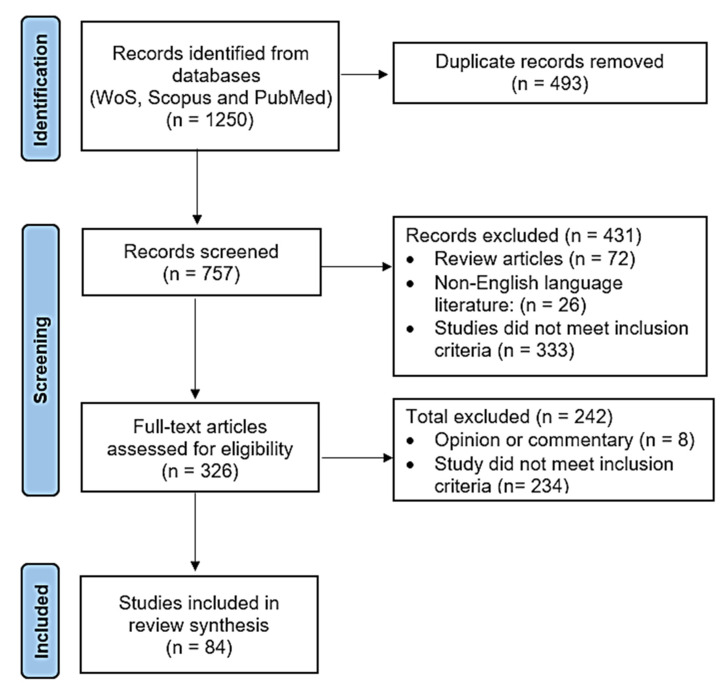
Preferred Reporting Items for Systematic Reviews and Meta-Analyses (PRISMA) flowchart showing the search strategy and selection process for the research articles used in the study.

**Figure 3 animals-13-01830-f003:**
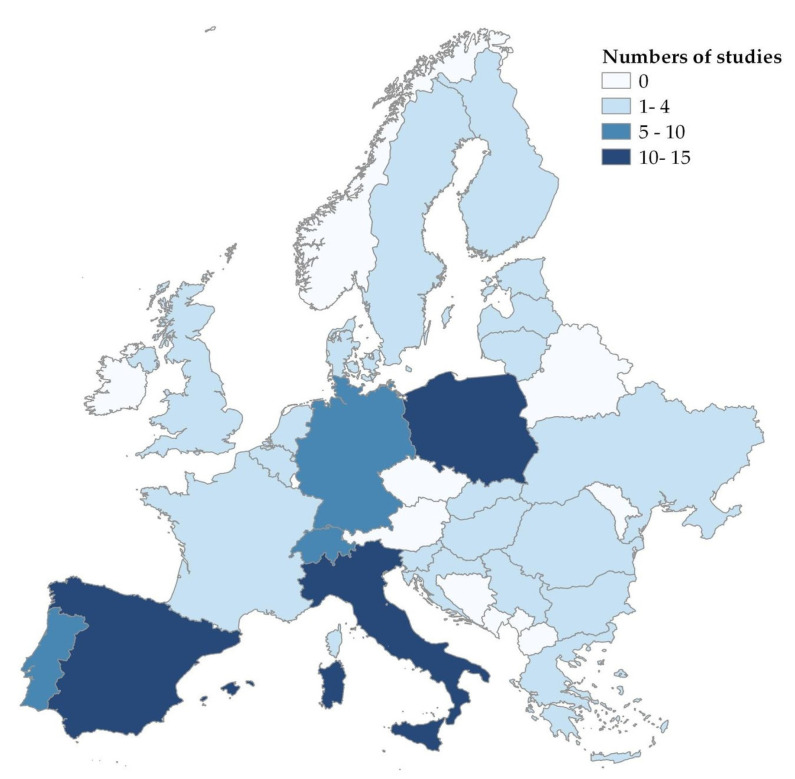
Country-wise distribution of the studies included in the systematic review. (Note: If a publication included multiple countries, it was counted once for each country included).

**Figure 4 animals-13-01830-f004:**
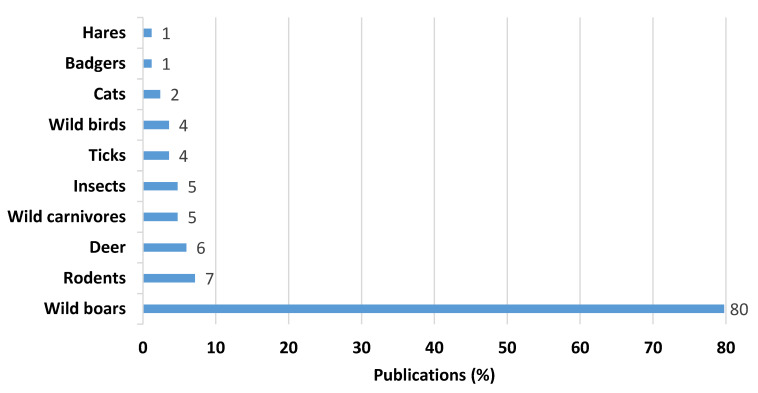
Distribution of studies based on the type of wildlife involved in the pathogen transmission to domestic pigs.

**Figure 5 animals-13-01830-f005:**
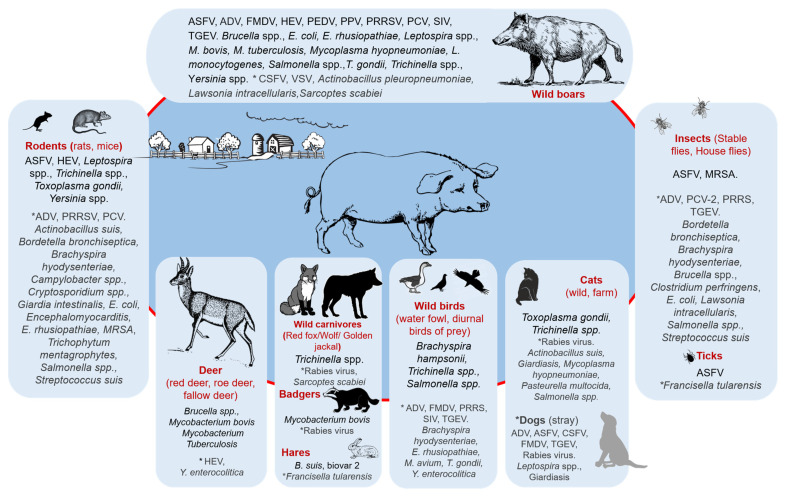
Pathogens associated with the various wildlife and pests linked to infections in domestic pigs in Europe. (* Pathogenic agents (or diseases) in light gray were included from studies not following the inclusion criteria and studies carried out outside Europe to highlight the potential of each host in the transmission of these pathogenic agents).

**Figure 6 animals-13-01830-f006:**
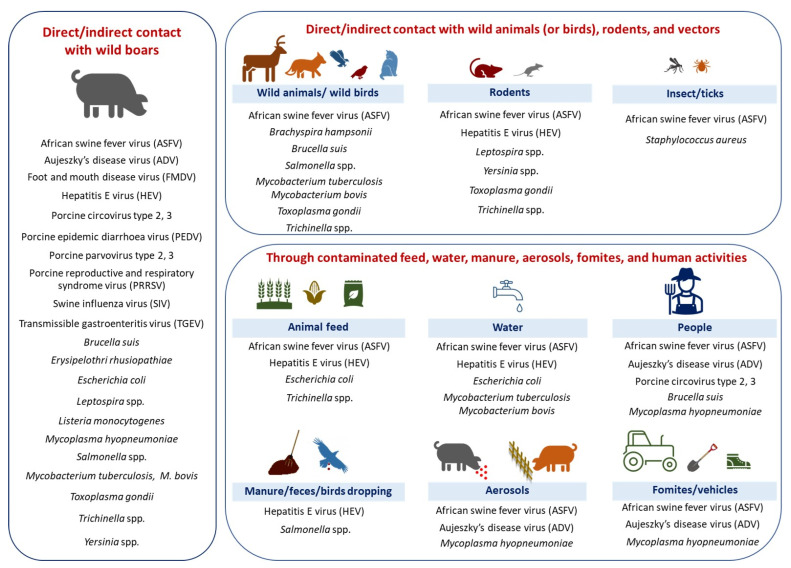
Direct and indirect pathogen transmission pathways from wildlife and pests to domestic pigs. (Note: In the present figure, we included the information from the selected studies; however, there could be other possible transmission pathways.)

**Table 1 animals-13-01830-t001:** The inclusion and exclusion criteria adopted in the literature review process.

Sl. No.	Inclusion Criteria	Exclusion Criteria
1.	Peer-reviewed, original research articles from the studies carried out in the European region	A manuscript that was not peer-reviewed and outside the European region
2.	Written in the English language	A manuscript not written in the English language
3.	The study involved domestic pigs and wildlife species, pests	Studies carried out only among domestic pigs
4.	Study collected or analyzed data ‘to investigate the interface/interaction between wild animals and domestic pigs’ OR ‘the prevalence of common domestic pig pathogen(s) in wildlife and mention possible transmission risk factors’	Conference abstracts, review articles, commentary, and gray literature
5.	The study included information about the role of wild animals in spreading pathogenic agents on pig farms	The study did not involve pathogenic agents of domestic pigs
6.	The study described possible transmission routes of pathogenic agents from wildlife species to domestic pigs	The study focused exclusively on either wild pork products or domestic pork products
7.	The study described information relating to the risks of transmission of pathogenic agents from wildlife species to domestic pigs	The study on the livestock–wildlife interface was based only on the results from simulation models

**Table 2 animals-13-01830-t002:** A summary of the studies on the epidemiological role of wildlife and pests in the transmission of pathogenic agents to domestic pigs in European countries.

Country	Pathogen/ Disease	Prevalence in Domestic Pig	Source	Epidemiological Role	Prevalence in Wildlife	Evidence of Transmission Based on	Reference
Belgium	African swine fever virus (ASFV)	NA	Wild boars	Reservoir	NA	Expert opinion	[27]
Bulgaria	Foot-and-mouth disease virus (FMDV)	NA	Wild boars	Reservoir	NA	Authors’ hypothesis	[28]
	Hepatitis E virus (HEV)	60% (260/433)	Wild boars	Reservoir	12.5% (4/32)	Observation	[29]
Croatia	HEV	24.5%	Wild boars	Reservoir	12.3%	Observation	[30]
		NA	Rodents (Yellow-necked mice)	Potential reservoirs or/and transmitters	0.21% (1/483)	Observation	[31]
		15.2% (216/1419)	Wild boars	Reservoir	11.5% (83/720)	Authors’ hypothesis	[32]
Denmark	ASFV	NA	Insects (stable flies)	Mechanical vectors	NA	Authors’ hypothesis	[33]
		NA	Insects (stable flies)	Mechanical vectors	NA	Authors’ hypothesis	[34]
	Methicillin-resistant *Staphylococcus aureus* (MRSA)	NA	Insects (stable flies)	Mechanical vectors	5.4%	Observation	[35]
		NA	Insects (house flies)	Mechanical vectors	7.8%	Observation	
	*Toxoplasma gondii*	NA	Rodents (mice (*Mus musculus*))	Reservoir	8% (11/137)	Observation	[36]
Estonia	ASFV	NA	Wild boars	Reservoir	NA	Authors’ hypothesis	[37]
		29.7%	Wild boars	Reservoir	NA	Authors’ hypothesis	[38]
Finland	HEV	NA	Wild boars	Reservoir	18% (32/181)	Observation	[39]
	*Salmonella* spp.	NA		Reservoir	38% (69/181)	Observation	
	*T. gondii*	NA		Reservoir	9% (17/181)	Observation	
	*Trichinella* spp.	NA		Reservoir	1% (2/181)	Observation	
	*Brucella* spp.	NA		Reservoir	9% (8/87)	Observation	
	*Yersinia* spp.	NA		Reservoir	56% (102/181)	Observation	
France (Corsica) Corsica	*Escherichia coli* *(E. coli)*	NA	Wild boars	Reservoir	87.5% (7/8)	Observation	[40]
		NA	Wild boars	Reservoir	NA	Expert opinion	[41]
	HEV	88%	Wild boars	Reservoir	29.2% (101/346)	Genetic typing	[25]
	*Brucella suis*, biovar 2	104 isolates	Wild boars	Reservoir	45 isolates	Genetic typing	[26]
		104 isolates	European hares	Reservoir	54 isolates	Genetic typing	
Germany	Aujeszky’s disease virus (ADV)	NA	Wild boars	Potential host	6.64% (6795/ 102,387)	Observation	[42]
	*B. suis*, biovar 2	3 isolates	Wild boars	Reservoir	22 isolates	Genetic typing	[26]
		3 isolates	European hares	Reservoir	1 isolate	Genetic typing	
	HEV 3	NA	Wild boars	Reservoir and transmission host	16.8% (39/232)	Observation	[43]
		100% (4/4)	Wild boars	Reservoir and transmission host	100% (4/4)	Observation	[44]
	*Salmonella choleraesuis*	NA	Wild boars	Reservoir	84.7% (39/46)	Observation	[45]
Greece	ADV	NA	Wild boars	Reservoir	32.0% (115/359)	Observation	[46]
	*Erysipelothrix rhusiopathiae*	NA		Reservoir	2.4% (4/167)	Observation	
	PRRSV	NA		Reservoir	5.6% (18/321)	Observation	
	*Mycoplasma hyopneumoniae*	NA		Reservoir	72.55% (229/316)	Observation	
Hungary	*B. suis*, biovar 2	8 isolates	Wild boars	Reservoir	55 isolates	Genetic typing	[26]
		8 isolates	European hares	Reservoir	5 isolates	Genetic typing	
Italy	*B. suis*, biovar 2	NA	Wild boars	Reservoir	1.3% (5/389)	Observation	[47]
		9 isolates	Wild boars	Reservoir	114 isolates	Genetic typing	[26]
		11 strains	Wild boars	Reservoir	98.3% (170/173 strains)	Observation	[48]
	*B. melitensis*, biovar 3	NA	Wild boars	Reservoir	1.7% (3/173)	Observation	
	HEV	NA	Rodents [black rats (*Rattus rattus*)]	Reservoir	2% (1/47)	Observation	[49]
		NA	Wild boars	Reservoir	13.7% (40/291)	Authors’ hypothesis	[50]
	HEV-3	NA	Wild boars	Reservoir	NA	Authors’ hypothesis	[51]
	*Leptospira* spp.	NA	Wild boars	Potential sources	NA	Authors’ hypothesis	[52]
	*Mycobacterium bovis*	4.9% (330/6714)	Deer (fallow deer)	Reservoir	25.5% (12/47)	Observation	[53]
		4.9% (330/6714)	Wild boars	Reservoir	6.8% (19/280)	Observation	
	Porcine circovirus type 2 (PCV-2)	NA	Wild boars	Reservoir	47.30% (70/148)	Observation	[54]
	Porcine circovirus type 3 (PCV-3)	NA	Wild boars	Reservoir	49.32% (73/148)	Observation	
	PCV-2	60% (12/120)	Wild boars	Reservoir	22% (28/127)	Genetic typing	[24]
	Porcine epidemic diarrhea virus (PEDV)	NA	Wild boars	Reservoir	3.83% (17/444)	Observation	[55]
	Transmissible gastroenteritis virus (TGEV)	NA	Wild boars	Reservoir	0.67% (3/444)	Observation	
Latvia	ASFV	NA	Wild boars	Reservoir	NA	Authors’ hypothesis	[56]
		NA	Wild boars	Reservoir	NA	Authors’ hypothesis	[57]
Lithuania	ASFV	NA	Insects (*Stomoxys*)	Mechanical vector	10.7% (217/2035)	Observation	[58]
		NA	Wild boars	Reservoir	75% (2513/3352)	Observation	[59]
		NA	Wild boars	Reservoir	NA	Authors’ hypothesis	[60]
	Porcine reproductive and respiratory syndrome virus (PRRSV)	NA	Wild boars	Reservoir	8.66% (298/1597)	Observation	[61]
The Netherlands	*Brucella* spp.	NA	Wild boars	Reservoir	6.4% (131/2057)	Observation	[62]
	*Leptospira* spp.	NA	Rodents	Reservoir	5.3% (20/379)	Observation	[63]
	*T. gondii*	NA	Rodents	Reservoir	1.6% (5/312)	Observation	
Poland	ASFV	NA	Wild boars	Reservoir	4.6% (15,639/ 340,775)	Observation	[64]
		NA	Wild boars	Reservoir	14.2% (57/402)	Observation	[65]
		362 outbreaks (124,382 pigs)	Wild boars	Reservoir	5824 cases	Observation	[66]
	HEV	44.1% (63/143)	Wild boars	Reservoir	31% (90/290)	Observation	[67]
	*B. suis*, biovar 2	6 isolates	Wild boars	Reservoir	5 isolates	Genetic typing	[26]
		6 isolates	European hares	Reservoir	3 isolates	Genetic typing	
	PEDV	NA	Wild boars	Reservoir	3.2% (5/157)	Observation	[68]
	*T. gondii*	NA	Wild boars	Reservoir	37.7% (50/398)	Observation	[69]
		NA	Cats	Reservoir	NA	Observation	
	*Trichinella spiralis*, *T. britovi*, *T. nativa*	0.0002% (150/86,989,313)	Rodents (rats)	Reservoir	23.3% (21/90)	Observation	[70]
			Red foxes (*Vulpes vulpes*)	Reservoir	0.04% (71/1740)	Observation	
			Wild boars	Reservoir	0.34% (4690/ 1,389,865)	Observation	
Portugal	HEV	NA	Wild boars	Reservoir	14% (4/29)	Observation	[71]
	*B. suis*, biovar 2	NA	Wild boars	Reservoir	33–59.3%	Observation	[72]
		32 isolates	Wild boars	Reservoir	56 isolates	Genetic typing	[26]
	ASFV	NA	Ticks (*Ornithodoros erraticus*)	Vectors	28% (13/47) farms with ticks detected	Observation	[73]
		NA	Ticks (*O. erraticus*)	Reservoir	8.8% (3/34) farms with ticks detected	Observation	[74]
		NA	Ticks (*O. erraticus*)	Vectors	NA	Authors’ hypothesis	[75]
Romania	ASFV	69.40% (3942/5680)	Wild boars	Reservoir	30.6% (1738/5680)	Observation	[76]
		200 outbreaks farms	Wild boars	Reservoir	NA	Risk factors	[77]
	*B. suis*, biovar 2	27 isolates	Wild boars	Reservoir	NA	Genetic typing	[26]
Serbia	ASFV	9000 pigs	Wild boars	Reservoir	NA	Authors’ hypothesis	[78]
	*Brucella* spp.	NA	Wild boars	Natural hosts (and/or vectors) are likely reservoir	9.4%, (45/480)	Observation	[79]
	*T. spiralis*	NA	Wild cat	Reservoir	100% (1/1)	Observation	[80]
	*T. britovi*	NA	Red foxes	Reservoir	50% (2/4)	Observation	
	*T. spiralis*, *T. britovi*	NA	Golden jackals (*Canis aureus*)	Reservoir	100% (3/3)	Observation	
	*T. britovi*	NA	Wolves (*Canis lupus*)	Reservoir	100% (4/4)	Observation	
	*Trichinella* spp.	0.12% (344/282,960)	Wild boars	Reservoir	11.7% (11/94)	Observation	[81]
			Golden jackals	Reservoir	53.8% (7/13)	Observation	
			Red foxes	Reservoir	12.3% (7/57)	Observation	
			Wolves	Reservoir	100% (3/3)	Observation	
Slovakia	PCV-2	NA	Wild boars	Reservoir	43.8% (85/194)	Observation	[82]
	Porcine parvovirus type 3 (PPV-3)	NA	Wild boars	Reservoir	19.1% (37/194)	Observation	
	*T. pseudospiralis*	All negative (*n* = 1,843,464)	Wild boars	Reservoir	0.04%	Observation	[83]
			Red foxes	Reservoir	9.58%	Observation	
			Wild birds of prey	Reservoir	1.11%	Observation	
Slovenia	*Salmonella* spp.	50% (9/18) isolates	Wild boars	Reservoir	11.1% (2/18) isolates	Authors’ hypothesis	[84]
Spain	*Salmonella* spp.	NA	Wild boars	Reservoir	7.7% (81/1041)	Observation	[85]
	ASFV	NA	Wild boars	Reservoir	NA	Authors’ hypothesis	[86]
		NA	Ticks (*O. erraticus*)	Vectors	28% (13/47) farms with ticks detected	Observation	[73]
	ADV	1.7%	Wild boars	Reservoir	49.6 ± 2.4%	Authors’ hypothesis	[87]
		NA	Wild boars	Reservoir	80–100%	Authors’ hypothesis	[88]
		NA	Wild boars	Reservoir	NA	Authors’ hypothesis	[89]
	*B. suis*, biovar 2	NA	Wild boars	Reservoir	33–59.3%	Observation	[72]
		96 isolates	Wild boars	Reservoir	48 isolates	Genetic typing	[26]
		96 isolates	European hares	Reservoir	2 isolates	Genetic typing	
		NA	Wild boars	Reservoir	59.3% (121/204)	Observation	[90]
	*Brachyspira hampsonii*	NA	Wild birds (waterfowl)	Reservoir	NA	Authors’ hypothesis	[91]
	*M. bovis*	NA	Red deer	Reservoir	NA	Authors’ hypothesis	[92]
		0.7%	Wild boars	Reservoir	22.4%	Observation	[93]
		NA	Red deer	Reservoir	6.2%	Observation	
	*Mycobacterium tuberculosis* complex (MTC)	NA	Wild boars	Reservoir	NA	Authors’ hypothesis	[94]
		NA	Deer	Reservoir	NA	Authors’ hypothesis	
	PRRSV	1.25%	Wild boars	Reservoir	2.0% (7/294)	Observation	[95]
	*T. gondii*	24.3%	Cats	Reservoir	NA	Authors’ hypothesis	[96]
Sweden	*E. rhusiopathiae*	NA	Wild boars	Reservoir	17.5%	Observation	[97]
	*M. hyopneumoniae*	NA	Wild boars	Reservoir	24.8%	Observation	
	PCV-2	NA	Wild boars	Reservoir	99.0%	Observation	
	PPV	NA	Wild boars	Reservoir	78.0% (233/286)	Observation	
	Swine influenza virus (SIV)	NA	Wild boars	Reservoir	3.8% (11/286)	Observation	
	*T. gondii*	NA	Wild boars	Reservoir	28.6% (82/286)	Observation	
	*Y. enterocolitica*	18% (11/60)	Rodents	Reservoir	5% (9/190)	Observation	[98]
	*Y. pseudotuberculosis*	0% (0/60)	Rodents	Reservoir	0.5% (1/190)	Observation	
Switzerland	*E. coli*	NA	Wild boars	Reservoir	9% (14/153)	Observation	[99]
	*Salmonella* spp.	NA	Wild boars	Reservoir	12% (19/153)	Observation	
	*Y. enterocolitica*	NA	Wild boars	Reservoir	35% (53/153)	Observation	
	*L. monocytogenes*	NA	Wild boars	Reservoir	17% (26/153)	Observation	
	*Y. pseudotuberculosis*	NA	Wild boars	Reservoir	20% (30/153)	Observation	
	ASFV	NA	Wild boars	Reservoir	NA	Authors’ hypothesis	[100]
	*B. suis*, biovar 2	7 isolates	Wild boars	Reservoir	30 isolates	Genetic typing	[26]
		7 isolates	European hares	Reservoir	4 isolates	Genetic typing	
	*B. suis*, biovar 2	NA	Wild boars	Reservoir	35.8% (86/240)	Observation	[101]
	PRRSV	NA	Wild boars	Reservoir	0.43% (1/233)	Observation	
	*B. suis*	5.3% (17/322) contact cases	Wild boars	Reservoir	22.4%	Observation	[102]
	*M. hyopneumoniae*	NA	Wild boars	Reservoir	26.2% (256/978)	Expert opinion	[103]
		NA	Wild boars	More likely recipient rather than transmitter	22 cases diagnosed by real-time PCR	Observation	[104]
	*Salmonella* spp.	NA	Wild boars	Reservoir	17% (21/126)	Observation	[105]
	*T. gondii*	NA	Wild boars	Reservoir	35% (44/126)	Observation	
United Kingdom (UK)	*Salmonella* spp.	80.8% (97/120)	Wild birds	Reservoir	84% (28/33)	Observation	[106]
	*M. bovis*	12.8% (112/874)	Wild boars	Host	NA	Authors’ hypothesis	[107]
		12.8% (112/874)	Badgers	Host	NA	Authors’ hypothesis	
Ukraine	PCV-2	NA	Wild boars	Reservoir	NA	Authors’ hypothesis	[108]

## Data Availability

Data sharing not applicable.

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
