# Peer review of "The Role of Wildlife and Pests in the Transmission of Pathogenic Agents to Domestic Pigs: A Systematic Review"

_animals, 2023, doi:10.3390/ani13111830_

Round 1
Reviewer 1 Report
Authors have realised a review about the role of wildlife and Pests in the tranmission of pathogenic agents to domestic pigs.
The manuscript is well written, the outcomes have useful applications and the topic is very interesting. Some small additions can be included. If authors will follow the suggestions given, I certainly recommend this case report for the pubblication.
Firstly, I suggest you to include in your study also sarcoptic mange because this disease is very important to report not only in wild animals but also in domestic one. Infact, the mite is very species specific as demonstrated also in recent study and it can be carried by wild boars in outdoor pig farms. It's also associated with the presence of other wild animals as for example foxes, that are sympatric animals.
For these reasons, I strongly advice you to include these works :
-https://doi.org/10.3390/life13040987
-https://doi.org/10.1186/s12917-018-1430-3
Then, I also advice you to speak about the development of recent technologies that can help scientific researchers to monitoring animals'diseases. In particular, GIS and Remote Sensing can be used to study epidemiology of different zoonoses in a One Health approach.
Author Response
Comment: Authors have realised a review about the role of wildlife and Pests in the tranmission of pathogenic agents to domestic pigs. The manuscript is well written, the outcomes have useful applications and the topic is very interesting. Some small additions can be included. If authors will follow the suggestions given, I certainly recommend this case report for the pubblication. Firstly, I suggest you to include in your study also sarcoptic mange because this disease is very important to report not only in wild animals but also in domestic one. Infact, the mite is very species specific as demonstrated also in recent study and it can be carried by wild boars in outdoor pig farms. It's also associated with the presence of other wild animals as for example foxes, that are sympatric animals. For these reasons, I strongly advice you to include these works :
-https://doi.org/10.3390/life13040987
-https://doi.org/10.1186/s12917-018-1430-3
Response: Thank you for sparing your valuable time to review our manuscript and for providing positive feedback about our manuscript. We are pleased to hear that you found the manuscript both interesting and useful. Thank you for suggesting the inclusion of studies on sarcoptic mange. We have incorporated sarcoptic mange alongside with other pathogens transmitted by wild boars and wild carnivores (including foxes) to domestic pigs in Figure 5.
Additionally, we have cited both studies in the discussion chapter (references â„– 117, 119; Lines: 362, 364) to provide a comprehensive overview of different pathogens transmitted from wildlife to domestic pigs. We believe that this information will be beneficial for readers seeking a general understanding of this topic.
Comment: Then, I also advice you to speak about the development of recent technologies that can help scientific researchers to monitoring animals' diseases. In particular, GIS and Remote Sensing can be used to study epidemiology of different zoonoses in a One Health approach.
Response: Thank you for your valuable feedback. Your suggestion to focus on the development of recent technologies, particularly GIS and Remote Sensing, was duly noted and discussed in the discussion chapter (Lines: 350-352).
Reviewer 2 Report
Dear authors,
The review is organised with a clear research question. The word combinations used for the search were appropriately selected, and clear inclusion/exclusion criteria.
Please see specific suggestions below:
-Table 2: check if the formatting is right
-Table 2: Correct Reservoir[-]
-You report raw values for prevalence for some cells and not for others. Why?
-I believe you can remove the () in [(based on)]
-In Stomoxys, () are in italic, remove
-Choose one [Expert opinion] or [Expert opinions]
Figure 4. This figure could be half a page, and probably could be a bar plot.
Figure 5. Well done, but it can be improved. Maybe make the text in grey a slighter darker grey for improving contrast.
Table 3. This table makes it hard to memorize or recognize references. Not sure it should be in the main text. Maybe it could be a figure and the Table could be in Supplements?
-Check if the references are right.
Discussion
-L318: Do you believe maybe a lot of research or data gathered on these pathways is not published? I say that especially thinking about the low values for genetic typing.
-L346: Correct [spill over] for [spillover]
-L370: Is there an 'average' or gold standard document for farm biosecurity protocols common to Europe that you could discuss regarding good practices to be secured and applided based on your findings?
-L379: check spacing
No issues
Author Response
Point 1: Dear authors,
The review is organised with a clear research question. The word combinations used for the search were appropriately selected, and clear inclusion/exclusion criteria.
Response 1:
Thank you for your positive feedback on our manuscript. We are pleased to hear that you found the review well-organized, particularly with regard to the research question and the adopted methodology. Your positive assessment in these areas is greatly appreciated.
Point 2: Please see specific suggestions below: - Table 2: check if the formatting is right; -Table 2: Correct Reservoir[-]
Response 2: We have now corrected the formatting of Table 2.
Point 3: -You report raw values for prevalence for some cells and not for others. Why?
Response 3: Thank you for bringing this up. The reason behind the variation in reporting raw prevalence values across different cells in the table is due to the nature of the available studies. In some cases, certain studies did not provide explicit prevalence data, particularly in instances of outbreak investigations. However, we included these studies in our analysis based on our predefined inclusion and exclusion criteria. Despite the absence of raw prevalence values in some cells, these studies presented other relevant evidence and transmission analyses that we deemed necessary for our readers' understanding.
Point 4: -I believe you can remove the () in [(based on)]
Response 4: The correction has been done in Table 2
Point 5: -In Stomoxys, () are in italic, remove
Response 5: The correction has been done in Table 2
Point 6: -Choose one [Expert opinion] or [Expert opinions]
Response 6: The correction has been done in Table 2
Point 7: Figure 4. This figure could be half a page, and probably could be a bar plot
Response 7: Thank you for your valuable feedback. We have taken your suggestion and replaced Figure 4 in the bar plot.
Point 8: Figure 5. Well done, but it can be improved. Maybe make the text in grey a slighter darker grey for improving contrast.
Response 8: Thank you for bringing this to our attention. We have made the necessary edits to the figure as per your recommendation.
Point 9: Table 3. This table makes it hard to memorize or recognize references. Not sure it should be in the main text. Maybe it could be a figure and the Table could be in Supplements?
Response 9: Thank you for pointing out this relevant matter. We decided to shift the table to the Supplementary Materials as Table S3 and created figure (Figure 6) (Lines 206-207). In addition, we have also revised and modified Figure 3 into the map to make it more representative (Lines 154-155).
Point 10: -Check if the references are right.
Response 10: Some of the required corrections in the reference section have been done.
Discussion
Point 11:
-L318: Do you believe maybe a lot of research or data gathered on these pathways is not published? I say that especially thinking about the low values for genetic typing.
Response 11: Thank you for the comments. In our opinion conducting a comprehensive study simultaneously investigating pathogenic agents with genetic typing during outbreaks in wildlife and domestic pigs is highly challenging. It requires intensive coordination, resources, and logistical support. Coordinating surveillance across various wildlife populations and domestic pig farms, particularly during outbreaks, is complex and time-consuming. Additionally, managing sample collection, analysis, data, and ethical considerations adds to the overall complexity. These challenges can explain the limited number of publications in this field.
Point 12: L346: Correct [spill over] for [spillover]
Response 12: The correction has been done
Point 13: L370: Is there an 'average' or gold standard document for farm biosecurity protocols common to Europe that you could discuss regarding good practices to be secured and applided based on your findings?
Response 13: Thank you for bringing this to our attention. Farm biosecurity protocols in Europe vary across countries and regions, as there are no standardized gold standard documents. Each country typically has its own national guidelines or codes of practice for farm animal biosecurity. However, these protocols generally prioritize disease prevention, animal health protection, and the implementation of measures to minimize disease risks. In addition, in Chapter 12 of book “Biosecurity in Animal Production and Veterinary Medicine” (Dewulf et al., 2018 (reference â„– 124)) a good application of all components of biosecurity in pig production is described. Only in the case of specific diseases (e.g. ASF) in Europe there is common legislation according to the biosecurity (e.g. ASF -https://food.ec.europa.eu/african-swine-fever-latest-developments_en). At the WOAH currently a working group of experts, including one of the co-authors of this publication) are working on a new chapter for the terrestrial animal health code on biosecurity.
Point 14: L379: check spacing
Response 14: The correction has been done